# T1-Weighted Contrast Enhancement, Apparent Diffusion Coefficient, and Cerebral-Blood-Volume Changes after Glioblastoma Resection: MRI within 48 Hours vs. beyond 48 Hours

Davide Negroni [1,*], Romina Bono [1], Eleonora Soligo [2], Vittorio Longo [1], Christian Cossandi [3], Alessandro Carriero [1] and Alessandro Stecco [1] 

1 Radiology Department, Maggiore della Carità Hospital of Novara, 28100 Novara, Italy
2 Radiology Department, San Andrea Hospital of Vercelli, 13100 Vercelli, Italy
3 Neurosurgery Department, Maggiore della Carità Hospital of Novara, 28100 Novara, Italy
* Correspondence: dvdngr@gmail.com

**Abstract:** Background: The aim of the study is to identify the advantages, if any, of post-operative MRIs performed at 48 h compared to MRIs performed after 48 h in glioblastoma surgery. Materials and Methods: To assess the presence of a residual tumor, the T1-weighted Contrast Enhancement (CE), Apparent Diffusion Coefficient (ADC), and Cerebral Blood Volume (rCBV) in the proximity of the surgical cavity were considered. The rCBV ratio was calculated by comparing the rCBV with the contralateral normal white matter. After the blind image examinations by the two radiologists, the patients were divided into two groups according to time window after surgery: ≤48 h (group 1) and >48 h (group 2). Results: A total of 145 patients were enrolled; at the 6-month follow-up MRI, disease recurrence was 89.9% (125/139), with a mean patient survival of 8.5 months (SD 7.8). The mean ADC and rCBV ratio values presented statistical differences between the two groups ($p < 0.05$). Of these 40 patients in whom an ADC value was not obtained, the rCBV values could not be calculated in 52.5% (21/40) due to artifacts ($p < 0.05$). Conclusion: The study showed differences in CE, rCBV, and ADC values between the groups of patients undergoing MRIs before and after 48 h. An MRI performed within 48 h may increase the ability of detecting GBM by the perfusion technique with the calculation of the rCBV ratio.

**Keywords:** glioblastoma; MRI; surgical margins; diffusion; apparent diffusion coefficient; perfusion; dynamic susceptibility contrast; cerebral blood volume; T1 post-contrast; time window



## 1. Introduction

Gliomas are the most common form of brain tumor in adults, with an annual incidence of about 4 to 5 per 100,000 people [1–3]. In adults aged 45 to 75 years, the most common form is malignant glioblastoma (GBM). GBM accounts for 82% of cases of malignant glioma and is characterized histologically by considerable cellularity and mitotic activity, vascular proliferation, and necrosis [2]. The main treatment of GBM is total surgical resection [4].

The Radiologist is usually asked to evaluate the operative margins in the immediate post-surgical time [4,5]. The presence of a radiologically visible disease residual correlates with poorer patient survival and may require reoperation in selected cases [6–9].

The evaluation of the surgical cord is performed by T1-weighted, T2-weighted, DWI with ADC maps, and post-contrast T1-weighted sequences. The latter sequences are critical to discriminate between methemoglobin and spontaneously hyperintense in T1-weighted from tumor remnant due to Contrast Enhancement (CE) [7,10–12]. Studies from the 1990s argue that the appropriate diagnostic window is 3 days; after this time window, the appearance of surgically induced changes may hinder the diagnosis of residual tumors [13–15].

More recent studies have attempted to define the most appropriate time window by investigating changes in CE both before and after 72 h using high-field MRI (greater than 1.5 T) [10,16–18].

A recent systematic review found that CE changes may be best evaluated between 24 and 48 h after surgery, confirming the Updated Response Assessment Criteria for High-Grade Gliomas [18,19].

Only in recent years has attention been paid to Dynamic Susceptibility Contrast (DSC) sequences [19–22]. In pre-operative identification, the sensitivity and specificity of DSC appear to be high, at 94.7% and 93.7%, respectively, by the calculation of Cerebral Blood Volume (CBV) [23].

Diffusion (or DWI) and perfusion (or PWI) studies are becoming increasingly important. Given their high sensitivity values, they can play a key role in the post-surgical diagnosis of GBM.

However, studies describing brain post-operative changes in ADC and CBV values between MRIs performed before and after 48 h are not numerous in the literature. In the first 48 h after neurosurgery, the behavior of surgical margins in PWI is still unclear.

The aim of this study is to verify any differences in contrast enhancement, ADC, and CBV values obtained by DSC between post-operative MRI performed before and after 48 h.

## 2. Materials and Methods

### 2.1. Patients

From January 2014 to November 2021, every patient (N = 145) with a histopathological diagnosis of GGBM, according to the 2016 WHO criteria, was retrospectively retrieved from a electronic database [3].

The inclusion criteria were:

(1) Patients who underwent either first-time, partial, or total surgery and had an MRI examination.
(2) Availability of pre-operative MRI studies performed with a 1.5 T scanner, within 30 days before the surgery.
(3) Availability of post-operative MRI studies performed with a 1.5 T scanner, no later than 90 h post-surgery.

The exclusion criteria were:

(1) Image quality adequate to accurately perform all image analysis (for example, patients with evident movement artifacts) (5 patients excluded).
(2) Glioblastoma multiforme (GBM) characterized by relevant contrast enhancement, evaluated on preoperative MRI (0 patients excluded).

We analyzed only MRIs from 1.5 T because this is the standard in our center for the post-operative follow-up in GBM patients.

The choice of performing MRI before or after 48 h was made by the neurosurgeons. This choice was not driven by specific criteria. Primarily, the availability of the scanner was the main factor, followed by the clinical habits of the surgeon.

Data on the methylation status of oxygen 6-methylguanine-DNA methyltransferase (MGMT) promoter methylation status and isocitrate dehydrogenase 1 (IDH1) and/or IDH2 mutation were collected to better stratify the glioblastomas. Survival data or mortality rates were not available in our study.

### 2.2. Image Protocol

All the imaging investigations were performed on a 1.5 T MR scanner (MRI-Philips, Achieva Stream 1.5T) using a 16-channel head coil. Before contrast, we acquired axial T1-weighted spin-echo images; axial, coronal, and sagittal T2-weighted turbo-spin-echo images; axial-fluid-attenuated inversion recovery (FLAIR) images; and 3D T1-weighted gradient echo and DWI with ADC map.

Specifically, we acquired axial T1-weighted (TE/TR 15 ms/450 ms, slice thickness 5 mm with 1 mm interslice distance, matrix size 256 × 256), axial T2-weighted fast spin-

echo (TE/TR 100 ms/1000 ms, slice thickness 5 mm with 1 mm interslice distance, matrix size 512 × 512, and FOV in AP 230), and FLAIR images (inversion time 2800 ms, TE/TR 140 ms/11,000 ms, slice thickness 5 mm with 1 mm interslice distance, NEX 2, matrix size 480 × 480). Diffusion-weighted images were collected with TE/TR 90 ms/1000 ms (slice thickness 5 mm with 1 mm interslice distance, matrix size 320 × 320). ADC images were calculated from the acquired diffusion-weighted images with b 1000 s/mm$^2$ and b 0 s/mm$^2$ images.

Afterward, a dose of 0.1 mL/kg of Gadobutrol 1.0 mol/L (Gadovist, Bayer Schering Pharma, Berlin, Germany) was injected through a 20-gauge peripheral venous catheter via an automatic injector at a rate of 5 mL/s, followed by 20 mL at 5 mL/s of 0.9% saline.

After contrast infusion, we acquired fast field echo 3D axial T1-weighted images (sagittal: TE/TR 8/500 ms, slice thickness 1 mm with 0 interslice distance, matrix size 256 × 256), axial spin-echo T1-weighted images (TE/TR 15 ms/450 ms, slice thickness 5 mm with 1 mm interslice distance, matrix size 256 × 256), and axial Perfusion Weighted DSC images (PWI) (TE/TR 22/1778 ms, slice thickness 4 mm with 0 interslice distance, 22 slices, matrix size 256 × 256, EPI factor 53, no fold-over suppression, dynamic scan time 1sec.08). All the post-contrast imaging underwent a post-processing reconstruction. The software used was Philips Portal Intellispace 7.0 with "MRI Neuro Perfusion" tool (analysis: "leakage correction", map: rCBVcorr, mask: post-contrast fast field echo 3D axial T1-weighted or axial spin-echo T1-weighted images).

During the entire duration of the MRI scan, patients were instructed to keep their eyes closed and not to move their heads. No sedation or anesthesia was used in any of the patients.

Between 2014 and 2021, no substantial changes in the image protocols, hardware, or other routines were introduced in the post-operative glioblastoma MRI.

*2.3. Image Analysis*

MRI scans, conducted before and after surgery, were read by two independent neuro-radiologists (initials blinded for peer review with 4 years of post-residency experience and initials blinded for peer review with 5 years of post-residency experience) blinded to the degree of surgical resection (total or partial), the time between surgery and the post-op MRI, and related 6-month follow-up MRI scans. During the reporting session, both radiologists were able to evaluate the pre-operative MRI. Cases that presented a discordant opinion between the two radiologists were submitted to a third expert radiologist (initials blinded for peer review with more than 10 years of post-residency experience).

To assess the presence of residual tumors, CE areas in the proximity of the surgical cavity were considered. On T1-weighted imaging, the image pattern of CE near the cavity was divided into reactive CE and nodular CE. According to Majós et al. [24], reactive CE was considered as a stripe lining the surgical cavity due to surgery-related tissue damage (Figures 1 and 2). Nodular CE was evaluated as larger, inhomogeneous, solid micronodular masses around the cavity due to the persistence of neoplasia. CE areas and multiple lesions distant from the resection cavity as well as dural CE were not considered in this study [25].

The software used for the post-processing reconstruction was Philips IntelliSpace Portal 7.0 with the DSC reconstruction tool. Using the software, the perfusion maps were obtained.

The quantification of ADC and PWI values was obtained by designing a free-hand ROI at the parenchymal level of the surgical cavity. In particular, the slice was chosen based on the presence of CE in T1-weighted sequences and/or visible elevated perfusion on the rCBV map, paying attention to avoid the pore-brain cavity, clots, and hematic infarction areas [10]. All cases in which it was not possible to conduct this, either the ADC or PWI value was collected for the aforementioned reasons.

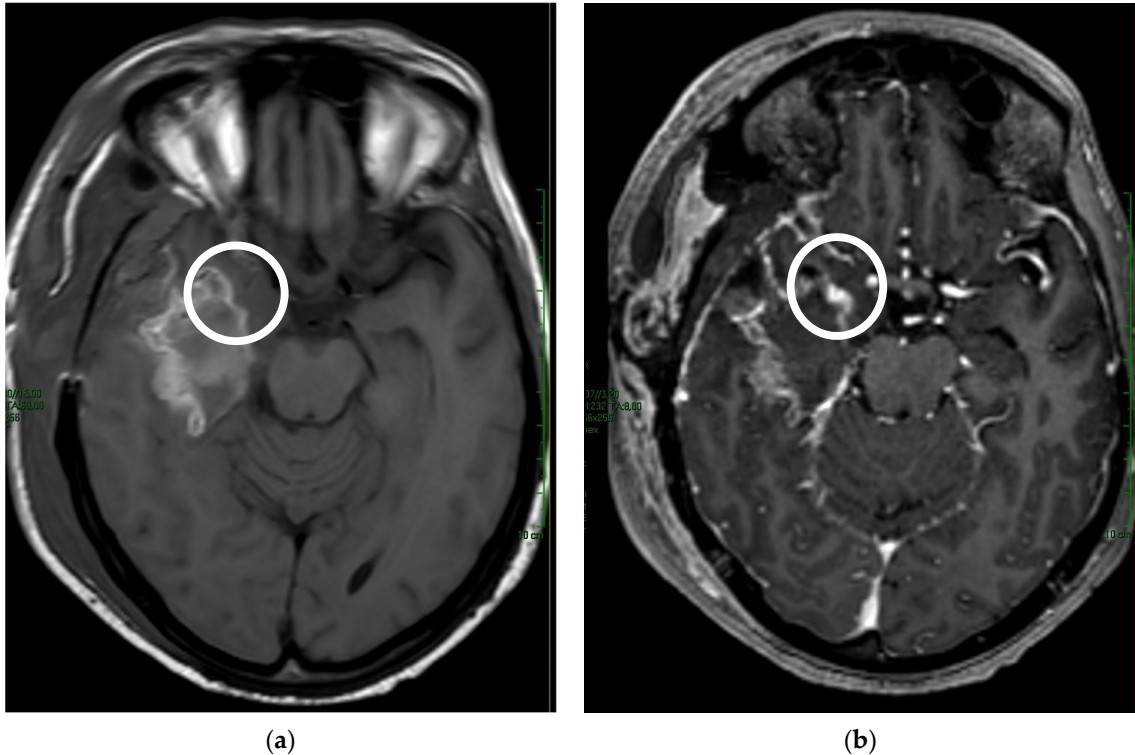

(**a**)                                                  (**b**)

**Figure 1.** Comparison of (**a**) T1-weighted pre-contrast with (**b**) T1-weighted post-contrast shows the nodular impregnation of the surgical margin (white circle).

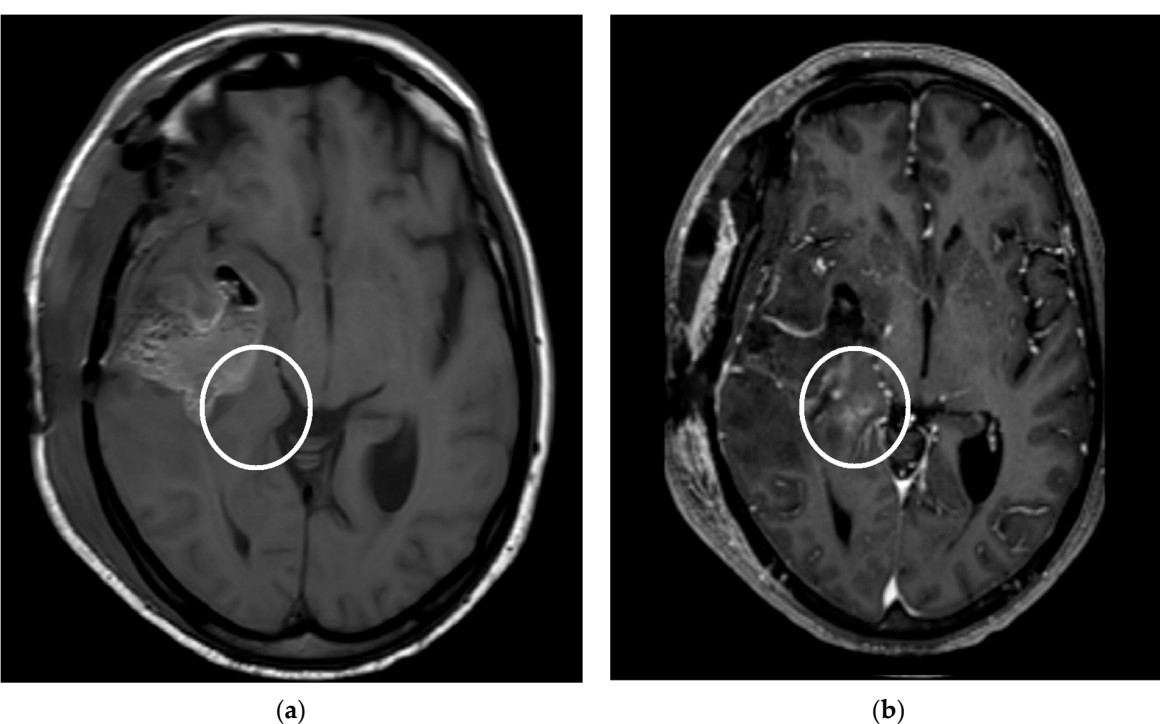

(**a**)                                                  (**b**)

**Figure 2.** Patient undergoing neurosurgical resection of glioblastoma multiforme in the right temporal site; MRI follow-up 40 h after surgery. (**a**) The T1-weighted image without contrast medium shows surgical sequel with pneumocephalus and inhomogeneous hyperintense signal at T1, with the possible expression of methemoglobin deposits. (**b**) The T1-weighted with contrast medium shows nodular contrast enhancement at the uncus of the parahippocampal gyrus; the match is compatible with residual glioblastoma multiforme (white circle).

The elevated perfusion seen visually in the resection area on the rCBV map was the region of interest (ROI) for measuring the rCBV [21,23]. The rCBV ratio was calculated by comparing the rCBV with a measurement of the contralateral normal white matter. An rCBV ratio of 1.0 was used, indicating a not elevated perfusion value. The software used for the post-processing reconstruction was Philips IntelliSpace 7.0 with MRI Neurology tool. According to the current literature, rCBV > 1.47 [26] and/or min ADC < 1.1 mm$^3$/s were considered indicative of post-operative GBM residue (Figure 3) [24,27–29].

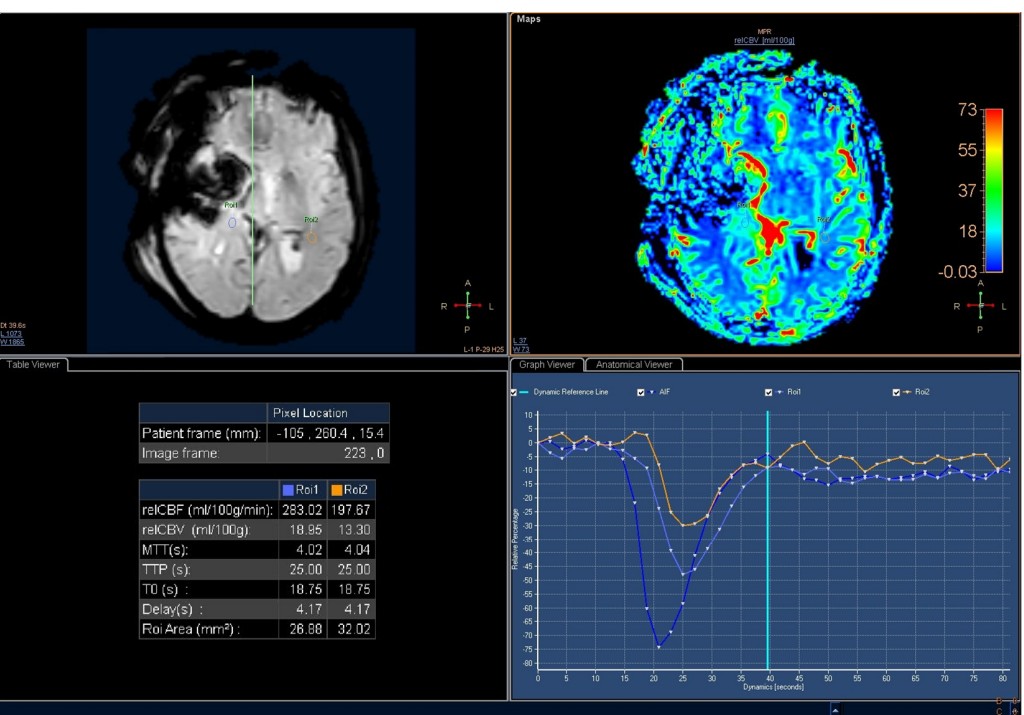

**Figure 3.** Perfusion Weighted Imaging of the same patient as in Figure 1. An increase in perfusion is visible ain the nodule (ROI 1) compared with the contralateral white matter (ROI 2). The rCBV ratio is greater than 1.4. ROI: Region Of Interest; AIF: Arterial Input Function.

When the neuroradiologists were not able to estimate an ADC or rCBV ratio, the value was classified as 'not estimable'.

After the blinded review of the images by the two Radiologists, patients were divided into 2 groups, according to the time window after surgery: ≤48 h and >48 h.

Since this is a retrospective study, the Ethical Board approval was waived. Nonetheless, every patient/next of kin signed a written informed consent allowing the anonymous use of their clinical records for research purposes.

### 2.4. Statistical Analysis

Statistical analysis was performed using STATA Software (StataCorp. 2015. Stata Statistical Software: Release 14. College Station, TX: StataCorp LP). A Fisher's exact test and Student's t-test were used to compare the different variables in the two groups (MRI ≤ 48 h and MRI > 48 h).

### 3. Results

Out of 145 patients who had a GBM diagnosis during the years of 2014–2021, a total of 139 patients (85 males and 54 females, mean age of 58.4 years, [SD 14.4]) fulfilled the inclusion criteria. A total of 75 out of 139 patients underwent a post-surgery MRI scan ≤48 h, while 64 out of 139 had it >48 h (Table 1).

**Table 1.** Main demographic data of the two groups. The first group had an MRI within 48 h, while the second group had an MRI after 48 h. A *p*-value < 0.05 was considered statistically significant.

|  | TOT | ≤48 h | >48 h | *p*-Value |
|---|---|---|---|---|
| Sample | 139 (100%) | 71 (51.1%) | 68 (48.9%) | *p* = 0.31 |
| Age (SD) | 59.2 (13.9) | 63.5 (11.4) | 54.6 (14.9) | *p* = 0.01 |
| Female | 46/139 (33.1%) | 27/71 (58.7%) | 19/68 (41.3%) | *p* = 0.11 |
| Male | 93/139 (61.2%) | 41/71 (44.1%) | 52/68 (55.9%) | |

No IDH1/2 mutations were found in our sample. MGMT promoter methylation was present in 57/139 patients (41.0%).

*Imaging Results*

A summary of the results is included in Table 2. All variables are in function of the two patient groups: post-operative MRI ≤ 48 h (group 1) and >48 h (group 2). The *p*-value is considered significant if <0.05.

**Table 2.** Main MRI findings were identified in the two study groups with their respective *p*-values. In the second section, the follow-up relapse data and the number of patients undergoing total resection of the glioblastoma were reported.

|  | TOT | ≤48 h | >48 h | *p*-Value |
|---|---|---|---|---|
| **MRIs** | 139 (100%) | 71 (56.0%) | 68 (44.0%) | |
| Reactive enhancement | 33/139 | 9 | 24 | *p* = 0.02 |
| Nodular enhancement | 106/139 | 62 | 44 | |
| **Ground Truth** | **TOT or Mean (SD)** | **≤48 h** | **>48 h** | ***p*-value** |
| 6-month MRI relapse | 125/139 | 60 | 65 | *p* = 0.12 |
| Gross total surgery | 104/139 | 55 | 54 | *p* = 0.41 |
| MGMT methylation | 57/139 | 32 | 25 | *p* = 0.26 |
| ADC estimable | 99/139 | 57 | 42 | |
| ADC Value | 0.90 (SD = 0.33) | 1.02 (SD = 0.36) | 0.74 (SD = 0.20) | *p* < 0.01 |
| ADC not estimable | 40/139 | 14 | 26 | *p* = 0.02 |
| rCBV estimable | 118/139 | 65 | 53 | |
| rCBV white matter | 147.8 (SD = 104.8) | 151.1 (SD = 110.3) | 143.3 (SD = 96.9) | *p* = 0.36 |
| rCBV ratio value | 2.31 (SD = 1.08) | 2.09 (SD = 0.79) | 2.56 (SD = 0.18) | *p* < 0.01 |
| rCBV not estimable | 21/139 | 6 | 15 | *p* = 0.03 |

*p* statistically significant at 0.05.

At the 6-month follow-up MRI, the total number of disease recurrences was 89.9% (125/139), with a mean patient survival of 8.5 (SD 7.8) months since the first diagnostic MRI was performed. The survival analysis showed no differences between group 1 and group 2 (*p* > 0.05).

In 79.9% of cases (111/139), the radiologists reported the persistence of disease at the first follow-up MRI, with no substantial differences between the two groups (*p* = 0.34). Of the remaining 20.1% (28/139 patients), i.e., in the 28 patients without an MRI and showing aspects of residual disease, GBM recurrence was found at the surgical cord level at 6 months after surgery in 25/28 patients (89.3%). In 8/111 patients (7.2%) in whom the persistent disease was reported, no disease recurrence was observed at 6 months (these patients received chemotherapy). These eight patients belonged to group 1, while no such cases were observed in group 2.

The mean ADC value was 0.90 mm$^3$/s (SD 0.33), with a median of 0.85; group 1 reported a mean ADC value of 1.02 mm$^3$/s (SD 0.36) with a median of 0.96, while group 2 reported a mean ADC value of 0.74 mm$^3$/s (SD 0.20) with a median of 0.70. The statistical analysis demonstrated a significant difference between group 1 and group 2, with $p < 0.01$.

In 40/139 patients, an ADC value could not be obtained due to the presence of artifacts because of hemoglobin residues. The analysis between the groups reported the superiority of these artifacts in MRIs performed 48 h after surgery, with $p = 0.02$.

Of these 40 patients in whom an ADC value was not obtained, rCBV values could not be calculated in 52.5% (21/40) due to artifacts. Again, the statistical analysis comparing groups 1 and 2 reported a significant $p$-value of 0.03.

## 4. Discussion

The study showed differences in terms of CE, rCBV, and ADC values between the group of patients who underwent MRI before and after 48 h.

Particularly, the ADC value was higher overall in patients in group 1 (post-operative MRI before 48 h) than in group 2 (post-operative MRI after 48 h).

This result could be related to both the presence of an ischemic-type insult present at the surgical margin and methemoglobin residues [10]. According to Strand et al., the presence of a peritumoral ischemic insult has been estimated to be 44% after diffuse glioma operation, with a greater prevalence at the level of the temporal lobe [30]. The difference in age between the two groups may have affected the figure. With increasing age, the surgically induced ischemic volume tends to increase [30].

Hence, as the hours pass beyond 48 h, the ADC value may decrease without indicating the presence of a residual GBM.

The analysis of the rCBV ratio values showed a significantly higher mean value in group 2 than in group 1.

Although the evaluation of rCBV has an important role in the diagnostic phase, there are still few data available in the literature about its use in the evaluation of surgical margins.

The result could be due to the hyperemic response of the margins that occurs 48 h after surgery. Both Bette et al. and Forsyth et al. identified an increase in reactive CE in the hours following surgery, indicative of a hypervascular response [25,31,32].

If the hypothesis were true, performing an MRI at 48 h could raise the sensitivity and specificity values of the perfusion analysis by rCBV ratio.

The number of cases in which ADC and rCBV could not be estimated was higher in group 2.

There is evidence, in the literature, of an increased occurrence of MRI changes caused by the surgical manipulation of the brain parenchyma [30,33,34]. Surgically induced changes become pronounced beyond 72 h [13,35–37], but may already be present in the first 48 h [13,33]. As in previous results, it seems that the more time passes after surgery, the more surgical artifacts are present in the surgical bed.

In the end, there was a significant difference between the classification of CE patterns: while group 1 showed superiority in terms of the presence of nodular CE, group 2 was superior for reactive CE.

The findings seem to agree with the current literature. Particularly, the study by Bette et al. identified a higher frequency of reactive CE beyond 45 h [25]. Moreover, Aliaga et al. demonstrated a residual overestimation in early post-surgical MRI, partially due to the misclassification of reactive CE [37]. An MRI within a time window of 48 h could lose precision in GBM identification based on CE.

A total of 8/139 patients (5.8%) were considered to be positive, although the control MRI at 6 months showed no signs of disease recurrence. All these patients belonged to group 1 and underwent radiotherapy and the same course of chemotherapy after the first post-operative MRI report. It should be noted that a "benign" nodular enhancement has been previously described but occurs infrequently (about 6.8% of all nodular enhancements) [25].

Unfortunately, it is not possible to establish the nature of the findings reported in this study (false positive MRI? Benign nodular enhancement? Residual tumor that responded to subsequent therapy?).

The absence of IDH1/2 mutations and the homogeneity distribution of MGMT promoter methylation in both groups of patients ($p > 0.05$) limit the bias associated with the GBM genotype. The prevalence of MGMT promoter methylation is in line with the current literature [9,38,39].

*Study Limitations*

The study had several limitations. Firstly, a statistically significant difference in age between the two groups is present. This possible confounding factor could be caused by a "selection bias" by our neurosurgeons: younger patients were more often candidates to have an MRI after 48 h, as per the protocol in force in the company, making this age group lower than the other arm. Secondly, we did not have a close follow-up MRI (MRIs within and after 48 h in the same patient), which could offer additional information about how the CE pattern, ADC value, and rCBV value change during this timeframe.

In this study, the same patients were not examined less than and more than 48 h after surgery; the patient groups were not fully comparable.

Data about confounding factors and radiation therapy, chemotherapy, and several complications after surgery were not available in this study. The absence of these variables limits the reliability of the results; however, the sample was not selected based on these factors, so the pre-and post-surgical treatment protocol was assumed to be homogeneous between groups.

## 5. Conclusions

An MRI performed within 48 h appears to improve the diagnostic value of Perfusion Weighted Imaging calculated by the DSC technique. In particular, the rCBV ratio becomes more accurate to identify a possible surgical residual glioblastoma. The presence of early ischemic events, induced by surgery, seems to limit the diagnostic capability of DWI and the corresponding ADC map in the first 48 h. Beyond 48 h, surgical manipulation-induced artifacts and blood may alter the detection of PWI. A time window within 48 h after surgery seems to be the most appropriate if a perfusion study is to be performed. This study has many limitations and further researches are needed to confirm the results. Future studies could aim to describe perfusion changes in the first 72 h by subjecting the same patients to multiple MRIs in order to avoid bias.

**Author Contributions:** Conceptualization, methodology, investigation, resources, formal analysis, data curation, D.N., E.S. and C.C.; writing—original draft preparation, E.S. and D.N. writing—review and editing, R.B. and V.L.; supervision, C.C., A.S. and A.C.; project administration, A.S. All authors have read and agreed to the published version of the manuscript.

**Funding:** This research received no external funding.

**Institutional Review Board Statement:** The study was conducted in accordance with the Declaration of Helsinki. Ethical review and approval were not required for this study in accordance with the national guidelines and institutional requirements.

**Informed Consent Statement:** Patient consent was waived due to the retrospective nature of this research.

**Data Availability Statement:** Not applicable.

**Acknowledgments:** Davide Negroni would like to thank Massimiliano Cernigliaro, Carolina Coda, Miriana Sassone, Agnese Siani, Sara Rodolfi, Riccardo Giuseppe Di Fiore, and Francesca Frattini for the moral support and useful advice during the conceptualization and text drafting.

**Conflicts of Interest:** The authors declare no conflict of interest.

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
