# Peer review of "T1-Weighted Contrast Enhancement, Apparent Diffusion Coefficient, and Cerebral-Blood-Volume Changes after Glioblastoma Resection: MRI within 48 Hours vs. beyond 48 Hours"

_tomography, doi:10.3390/tomography9010027_

Round 1

Reviewer 1 Report

The authors showed differences in CE, rCBV and ADC value between the group of patients undergoing MRI before 48 hours and after 48 hours after Glioblastoma resection. 

Please correct the following points

1 In the Background section of the Abstract, please indicate that it is about surgery for Glioblastoma.

2 The first letter of Background, Material and Methods in Abstract is not capitalized.

3 Please put the unit of Survival in line 24.

4 Isn't SDC in line 54 DSC?

5 It would be helpful for the readers to understand the MRI changes after Glioblastoma surgery, including the present results and previous reports, if you could provide a table showing which changes occur at what time after surgery.

Author Response

I thank the reviewers for their valuable advice and corrections. I personally find that the work has gained more clarity in content.
Below I have summarised the changes made in the text, following the point scheme proposed by the reviewers.
Reviewer 2.
1-5. I amended all the typos and miss.
R:“It would be helpful for the readers to understand the MRI changes after Glioblastoma surgery, including the present results and previous reports if you could provide a table showing which changes occur at what time after surgery”. 
Unfortunately, I am not able to provide this table in a short time. Since I do not have a graphics service, the creation of the table shown is very coarse and not very comprehensible. However, I have modified the conclusions to make the message of the study clearer.
CONCLUSION
MRI performed within 48 hours appears to improve the diagnostic value of Perfusion Weighted Imaging calculated by the DSC technique. In particular, the rCBV ratio becomes more accurate in identifying possible surgical residual Glioblastoma. The presence of early ischaemic events, induced by surgery, seems to limit the diagnostic capability of DWI and the corresponding ADC map in the first 48 hours. Beyond 48 hours, surgical manipulation-induced artifacts and blood may alter the detection of PWI. The time window within 48 hours after surgery seems to be the most appropriate if a perfusion study is to be performed. This study has many limitations and further research is needed to confirm the result. Future studies could aim to describe perfusion changes in the first 72 hours by subjecting the same patients to multiple MRIs in order to avoid bias.

Reviewer 2 Report

The authors attempted to identify any advantages of performing post-operative MRI less than 48 hours compared to a corresponding MRI examination carried out later than 48 hours. T1-weighted contrast enhancement, apparent diffusion coefficient and relative cerebral blood volume (rCBV) in the proximity of the surgical cavity were evaluated. The rCBV ratio was calculated by comparing the rCBV with the contralateral normal white matter. A total of 145 patients were divided into two groups, i.e., depending on MRI time window after surgery (≤ 48 hours [group 1], and > 48 hours [group 2]). The mean ADC value and the rCBV ratio showed statistically significant differences between the two groups. Considering that there have been recent attempts to assess the validity of old studies recommending an appropriate diagnostic window of 72 hours, this study is of some relevance. However, in the introduction, the authors need to clarify the added value of the present study compared with other previous studies. Furthermore, not the same patients were examined both less than and more than 48 hours after surgery, which might imply that the patient groups were not entirely comparable.

General comments

1. The paper is in need of a review of the English.

2. I would like the authors to clarify the relevance and added value of their study in the introduction.

3. Lines 75-77: The authors state the following: “The choice of performing an MRI before or after 48 hours was chosen by neurosurgery. The need for early imaging, the availability of the equipment, and factors associated with the operation were the main factors of choice by the specialist.” The factors “need for early imaging” and “factors associated with the operation” imply that these patients were selected for early imaging for a reason associated with their specific condition (related to the tumour or the surgery). This might mean that the results for these patients, in terms of imaging parameters, can not be expected to be same as for other patients. Hence, the observed differences in imaging parameters between groups may result from the fact that the conditions of the patients differed systematically, and not from the time of imaging. The selection criteria may thus have introduced a bias that complicates the interpretation of the findings.

4. Lines 84-95: Basic information about the employed MRI pulse-sequence and protocols needs to be added (TE, TR, b-values, etc.).

5. Sections 2.2-2.3: Basic information about rCBV calculation and image post-processing needs to be added. No preload was used, so how was contrast agent leakage corrected for in the rCBV calculation?

6. Lines 121-123: I am a little bit confused by the statements about the rCBV ratios. The rCBV ratio between normal grey matter (GM) and normal white matter is approximately 2, so if you had measured in normal GM, a ratio of 2.0 would have reflected “not elevated perfusion”. I am surprised that a ratio of 1.47 would be sufficiently high to identify GBM residue. I am not claiming that it is incorrect, but please give a specific reference (article and page). A ratio given with three significant digits seems to imply that it originates from one specific paper.

7. CBV in WM might show a certain decrease with age (perhaps up to 5% per decade), which may affect the rCBV ratio to some extent. As you already include rCBV in Table 2, it would be illustrative also to include the normal WM rCBV values, used for reference, in Table 2.

8. Line 63: Patients were included over a rather long time period (2014 to 2021), and the authors should clarify whether substantial changes in image protocols, hardware or other routines were introduced at the department in question during this period.

 Minor comments

9. Line 17: It is stated the aim was to compare MRI at 48 h with MRI at 72 h, but this was not done in this study. The authors compared ≤ 48 hours with > 48 hours, which is not the same thing.

10. Lines 39-40: How is “immediately” defined?

11. Lines 42-43 (and other places): “T1” should be “T1-weighted”

12. Line 53: SDC should be DSC

13. Lines 261-263: The conclusion is too brief and sends a rather contradictive message to the reader. I understand that it accurately reflects the results, but the authors’ opinion about whether or not to recommend MRI ≤ 48 hours could be made somewhat clearer to the readers.

14. Lines 261-263: It is, perhaps, also important to point out that if the time of imaging after surgery differs between patients, the results will not be comparable, i.e., the time of imaging needs to be considered in order to correctly interpret the findings.

Author Response

I thank the reviewers for their valuable advice and corrections. I personally find that the work has gained more clarity in content.

Below I have summarised the changes made in the text, following the point scheme proposed by the reviewers.

Reviewer 1.

  1. An extensive revision of English was carried out, correcting some verbal constructs and typos.
  2. Sono state aggiunte le seguenti frasi in introduzione per enfatizzare lo studio : Diffusion (or DWI) and Perfusion (or PWI) studies are gaining importance. Given their high sensitivity values, they can play a key role in the post-surgical diagnosis of Glioblastoma.

However, studies describing brain post-operative changes between before 48 and after 48 in ADC and CBV values are not numerous in the literature. In the first 48 hours after neurosurgery, the behavior of surgical margins in PWI is still unclear”

  1. In lines 75-77 I removed the sentence 'The need for early imaging, the availability of the equipment and factors associated with the operation were the main factors in the specialist's choice' and clarified the aspect of performing an MRI in an early window after consultation with neurosurgeons.

4 . In the section 'Image Protocol' I added the specifications for the execution of the individual sequences.

Perfusion maps were obtained using Philips Intellispace software.

  1. The rCBV cut-off of 1.47 was chosen in accordance with Kong et al. "Diagnostic Dilemma of Pseudoprogression in the Treatment of Newly Diagnosed Glioblastomas: The Role of Assessing Relative Cerebral Blood Flow Volume and Oxygen-6-Methylguanine-DNA Methyltransferase Promoter Methylation Status".
  2. For the calculation of the rCBV ratio, the WM of the contralateral hemisphere was used (taking care not to take any cortical turns). I have added in Table 2 the average of the WM CBV values used as a reference.
  3. Added sentence specifying the time period: 'Between 2014 and 2021, no substantial changes in imaging protocols, hardware or other routines in post-operative glioblastoma MRI'.

9-12. Small errors have been corrected according to the indications

13, 14. Conclusions have been rescripted with a clear indication of the authors’ opinion and attention to future studies.

Moreover, in the "study limitation" section, I added the following sentence. The same patients were not examined both less and more than 48 hours after surgery; the patient groups were not fully comparable.

Round 2

Reviewer 2 Report

DSC-MRI protocol and analysis details are still missing, e.g., TE, temporal resolution, number of slices, slice thickness, in-plane resolution, readout type (GRE or SE). How was potential contrast agent leakage corrected for in the post-processing?

Author Response

Thank you for the clarification. I have completed the following paragraph as requested.

2.2 Image Protocol

All the imaging investigations were performed on a 1.5 T MR scanner (MRI-Philips, Achieva Stream 1.5T) using a 16-channel head coil. Before contrast, we acquired axial T1-weighted spin-echo images, axial, coronal, and sagittal T2-weighted turbo-spin-echo images, axial fluid-attenuated inversion recovery (FLAIR) images, 3D T1-weighted gradient-echo and DWI with ADC map.

Specifically: the axial T1-weighted (TE/TR 15 ms/450 ms, slice thickness 5 mm with 1 mm interslice distance, matrix size 256 × 256); axial T2-weighted fast spin-echo (TE/TR 100 ms/1000 ms, slice thickness 5 mm with 1 mm interslice distance, matrix size 512 × 512, and FOV in AP 230); FLAIR images (inversion time 2800 ms, TE/TR 140 ms/11,000 ms, slice thickness 5 mm with 1 mm interslice distance, NEX 2, matrix size 480 × 480). Diffusion-weighted images were collected with TE/TR 90 ms/1000 ms, slice thickness 5 mm with 1 mm interslice distance, matrix size 320 × 320). ADC images were calculated from acquired diffusion-weighted images with b 1000 s/mm2 and b 0 s/mm2 images.

 After, a dose of 0.1 ml/kg of Gadobutrol 1.0 mol/l (Gadovist, Bayer Schering Pharma, Berlin, Germany) was injected through a 20-gauge peripheral venous catheter via an automatic injector at a rate of 5 ml/s, followed by 20 ml at 5 ml/s of 0,9% saline. 

 After contrast infusion, we acquired a fast field echo 3D axial T1-weighted images (sagittal: TE/TR 8/500 ms, slice thickness 1 mm with 0 interslice distance, matrix size 256 × 256), an axial spin-echo T1-weighted images (TE/TR 15 ms/450 ms, slice thickness 5 mm with 1 mm interslice distance, matrix size 256 × 256), and the axial Perfusion Weighted DSC imaging (PWI) (TE/TR 22/1778 ms, slice thickness 4 mm with 0 interslice distance, 22 slices, matrix size 256 × 256, EPI factor 53, no fold-over suppression, dynamic scan time 1sec.08 ). All the post-contrast imaging underwent a post-processing reconstruction. The software used was Philips Portal Intellispace 7.0 with “MRI Neuro Perfusion” tool (analysis: “leakage correction”, map: rCBVcorr, mask: post-contrast fast field echo 3D axial T1-weighted or axial spin-echo T1-weighted images).